behaviour, genetics, immunology

balancing selection, female fecundity, maternal immunization, homosexuality, older brothers, fraternal birth order

**Author for correspondence:**
Ray Blanchard
e-mail: ray.blanchard.phd@gmail.com

# A method yielding comparable estimates of the fraternal birth order and female fecundity effects in male homosexuality

Ray Blanchard[1], Jurian Krupp[2], Doug P. VanderLaan[3], Paul L. Vasey[4] and Kenneth J. Zucker[1]

[1]Department of Psychiatry, University of Toronto, Toronto, Ontario, Canada
[2]Department of Health and Human Sciences, Institute of Sexology and Sexual Medicine, Charité-Universitätsmedizin Berlin, Berlin, Germany
[3]Department of Psychology, University of Toronto Mississauga, Mississauga, Ontario, Canada
[4]Department of Psychology, University of Lethbridge, Lethbridge, Alberta, Canada

RB, 0000-0002-4499-3521; JK, 0000-0003-3669-1141; DPVL, 0000-0003-4498-9175; KJZ, 0000-0001-5313-6401

The fraternal birth order effect (FBOE) is the finding that older brothers increase the probability of homosexuality in later-born males, and the female fecundity effect (FFE) is the finding that the mothers of homosexual males produce more offspring than the mothers of heterosexual males. In a recent paper, Khovanova proposed a novel method for computing independent estimates of these effects on the same samples and expressing the magnitude and direction of the effects in the same metric. In her procedure, only families with one or two sons are examined, and daughters are ignored. The present study investigated the performance of Khovanova's method using archived data from 10 studies, comprising 14 samples totalling 5390 homosexual and heterosexual subjects. The effect estimate for the FBOE showed that an increase from zero older brothers to one older brother is associated with a 38% increase in the odds of homosexuality. By contrast, the effect estimate for the FFE showed that the increase from zero younger brothers to one younger brother is not associated with any increase in the odds of homosexuality. The former result supports the maternal immune hypothesis of male homosexuality; the latter result does not support the balancing selection hypothesis.

## 1. Introduction

The fraternal birth order effect (FBOE) is the finding that older brothers increase the probability of homosexuality in later-born males,[1] and the female fecundity effect (FFE) is the finding that the mothers of homosexual males produce more offspring than the mothers of heterosexual males.[2] There is a considerable amount of empirical evidence for the reproducibility of the FBOE [2–5] and the FFE [6–8]. Each effect relates to a specific biological theory of the aetiology of homosexuality in males—an immunological theory, in the case of the FBOE [9,10], and a genetic theory, in the case of the FFE [11].

The *maternal immune hypothesis* (MIH) is the hypothesis that the FBOE reflects the progressive immunization of some mothers to male-specific antigens and the consequent effects of anti-male antibodies on sexual differentiation in the brain in male fetuses. According to this hypothesis, cells (or cell fragments) from male fetuses enter the maternal circulation during childbirth or perhaps earlier in pregnancy. These cells include substances that occur primarily on the surfaces of male brain cells, for example, the Y-linked membrane proteins NLGN4Y and PCDH11Y. The mother's immune system recognizes these male-specific molecules as foreign and produces antibodies to them. In subsequent male pregnancies, her antibodies cross the placental barrier and enter the fetal

brain. Once in the brain, these antibodies bind to male-specific molecules on the surface of neurons and prevent these neurons from 'wiring-up' in a fully male-typical pattern. In consequence, the individual will later be attracted to men rather than to women. There has only been one laboratory test of the MIH, but its results were consistent with the hypothesis. This test found higher concentrations of anti-NLGN4Y antibody in the sera of mothers of homosexual men, especially those with older brothers, compared with the concentrations for mothers of heterosexual control subjects [10].

The FFE is a prediction of the *balancing selection hypothesis* (BSH). The BSH is an attempt to reconcile the findings from behaviour genetics and molecular genetics that homosexuality in men is partially heritable with the finding that homosexual men produce far fewer offspring than do their heterosexual counterparts. If both these findings are true—and there is no particular reason to doubt either—then the number of people who carry genes predisposing to homosexuality should be declining and the prevalence of homosexuality in the male population should be decreasing. Such a decrease, however, is not evident. The BSH resolves this seeming conundrum by proposing that the same genes that predispose to homosexuality in some males also increase fecundity in their heterosexual relatives, especially female relatives; this is the predicted FFE. Because of the FFE, the family's total number of descendants and the number of individuals carrying 'gay genes' remain constant. The FFE compensates for the low fertility of homosexual men.

We are not primarily focused, in this article, on the MIH or the BSH *per se*. These hypotheses are the underlying reason why methodological studies of the FBOE and FFE are interesting and important. We are primarily concerned here with the observable variables older brothers and family size, with their statistical entanglement, and with a novel methodology intended to produce a cleaner picture of the relation between each variable and male sexual orientation.

Khovanova [1] proved mathematically that the FBOE and FFE are inherently related. The FBOE implies a correlation between homosexuality and maternal fecundity. Conversely, the FFE implies a correlation between homosexuality and number of older brothers. As a solution to this confounding, Khovanova proposed a novel method for computing independent estimates of the FBOE and the FFE on the same samples and expressing the magnitude and direction of these effects in the same metric. In her procedure, only families with one or two sons are considered, and daughters are ignored.

Khovanova's procedures require three parameters calculated from an empirical sample: $p_{11}$ is the probability that the first (and only) boy in a one-son family is homosexual, $p_{12}$ is the probability that the first boy in a two-son family is homosexual and $p_{22}$ is the probability that the second boy in a two-son family is homosexual. Given these definitions, the FFE implies that $p_{12} > p_{11}$, and the FBOE implies that $p_{22} > p_{12}$. Thus, if both effects are present, then $p_{22} > p_{12} > p_{11}$.

These inequalities may be easier to understand without the mathematical notation. The logic behind the inequality $p_{22} > p_{12}$ is this: the second of two boys and the first of two boys have equally fecund mothers (two sons), but the second of two boys has a higher fraternal birth order (one older brother versus zero older brothers). Therefore, the second of two boys is more likely to be homosexual than the first of two boys.

The logic behind the inequality $p_{12} > p_{11}$ is this: the first of two boys and a first and only boy have the same fraternal birth order (zero older brothers), but the first of two boys has a more fecund mother (two sons versus one son). Therefore, the first of two boys is more likely to be homosexual than a first and only boy.

Khovanova expresses the magnitude and direction of both effects as risk ratios[3] (sometimes called relative risks), that is, ratios of probabilities. In her words, 'The ratio $p_{12}/p_{11}$ shows a contribution of FF independent of FBOE' and 'The ratio $p_{22}/p_{12}$ shows the contribution of FBOE independent of FF.' Another way to put this is that the ratio $p_{12}/p_{11}$ represents the multiplicative change in the probability of homosexuality associated with the increase from zero to one younger brother.[4] Similarly, the ratio $p_{22}/p_{12}$ represents the multiplicative change in the probability of homosexuality associated with the increase from zero to one older brother.

Khovanova's procedure is simple, logical and elegant. It is certainly more transparent than multivariate procedures for disentangling the effects of birth order and family size. It raises certain questions, however, that can be settled only through empirical investigation. First, Khovanova's procedure is a species of matching procedure, and like all matching procedures, it must entail some degree of data loss. Would enough cases remain from an 'average' size sample to make statistical significance for the results a realistic possibility?

Second, Khovanova's procedure is carried out on the earlier-born members of a sibship. How well do its quantitative results apply to the later-born members of a sibship? Do results obtained with Khovanova's method look generally similar to results obtained with traditional methods, which usually analyse the whole sibship? A similar question can be asked about the simplifying strategy of simply ignoring female offspring.

To answer these questions, we investigated the performance of Khovanova's method using archived data from 10 studies comprising 14 samples. With one exception, these studies were chosen for simple convenience. The exception [12] was deliberately selected because the original authors had detected both an FFE and an FBOE in the data. The first author chose eight studies simply because the data were conveniently in his computer files and ready for analysis. The first author had requested the 10th dataset, (J Krupp 2014, unpublished manuscript), well before the publication of Khovanova [1] and for a different purpose, but it happened to arrive while analysis for the present study was ongoing.

## 2. Material and methods

### (a) Subjects

Table 1 summarizes the major characteristics of the subjects from the 10 studies (e.g. country of residence, research volunteer or clinical patient). It also shows the number of subjects with complete data and the (always lower) number who came from one-son or two-son families. Additional demographic information (mean age, education and so on) may be found in the original publications.

The unpublished raw data from J Krupp 2014, unpublished manuscript will be described in more detail. This dataset contains sibship information on 401 help-seeking, self-referred males who participated in a first clinical interview for the sexual offense prevention outreach programme Project Dunkelfeld [24]. The number of cases with required information for present purposes was 388. These had a mean age of 37.63 years (s.d. = 12.11). About half the sample, 52%, had 10 years of education or less, and 48% had 11 years or more.

**Table 1.** Studies in the meta-analyses.

| authors | description of the sample | N complete sibling data | N 1- and 2-son families | per cent data loss |
|---|---|---|---|---|
| Blanchard & Bogaert [13] | Canadian volunteers | 736 | 507 | 31 |
| Blanchard et al. (B-NB) [14] | Canadian men (e.g. adoptees) reared in environments other than biological families: 'Bogaert (non-biological families)' subsample | 502 | 386 | 23 |
| Blanchard et al. (B-O) [14] | Canadian homosexual community volunteers and heterosexual university students: 'Bogaert (Other)' subsample | 415 | 295 | 29 |
| Blanchard et al. (H) [15] | Canadian patients referred to a specialty clinic, phallometrically diagnosed as homosexual or heterosexual hebephiles | 783 | 436 | 44 |
| Blanchard et al. (P) [15] | Canadian patients referred to a specialty clinic, phallometrically diagnosed as homosexual or heterosexual paedophiles | 242 | 147 | 39 |
| Blanchard et al. (T) [15] | Canadian patients referred to a specialty clinic, phallometrically diagnosed as homosexual or heterosexual teleiophiles | 1089 | 628 | 42 |
| Blanchard et al. [16] | British and American volunteers, from earlier studies by Siegelman [17–21] | 610 | 477 | 22 |
| Ellis & Blanchard [22] | American and Canadian volunteers | 1146 | 836 | 27 |
| Khorashad et al. [23] | Iranian homosexual male-to-female transsexuals and heterosexual cissexual psychiatric patients | 164 | 53 | 68 |
| Krupp (H) (J Krupp 2014, unpublished manuscript) | German patients self-referred for homosexual or heterosexual hebephilia or hebeteleiophilia, from the study by Beier et al. [24] | 113 | 99 | 12 |
| Krupp (P) (J Krupp 2014, unpublished manuscript) | German patients self-referred for homosexual or heterosexual paedophilia or paedohebephilia, from study by Beier et al. [24] | 55 | 48 | 13 |
| Schagen et al. [25] | Dutch biologically male, peripubertal gender-dysphoric patients and presumably cissexual heterosexual adolescent controls | 969 | 834 | 14 |
| VanderLaan & Vasey [12] | Samoan transgender same-sex-attracted males (fa'afafine) and cisgender heterosexual males | 538 | 176 | 67 |
| VanderLaan et al. [26] | Canadian children and adolescents referred to a child and adolescent gender identity clinic | 556 | 468 | 16 |

Subjects reported, in clinical interviews, their erotic interests with regard to biological sex (male, female or both) and were accordingly classified as homosexual, heterosexual or bisexual. They also reported their erotic interests with regard to age (prepubertal, pubertal or physically mature persons) and could accordingly be classified as paedophilic, hebephilic or teleiophilic. Erotic age-preference was captured by three, partially overlapping variables, and there were more than three categories available for final classification. Thus, for example, a subject could be classified as a paedohebephile rather than a paedophile or hebephile if he indicated erotic interest in children who were either prepubertal or pubertal.

In the present study, we excluded self-described bisexuals, because they did not fit into the research design and because we had no hypothesis about them, and we excluded teleiophiles, because hardly any of the teleiophiles were homosexual. We also excluded subjects who described themselves as attracted both to prepubertal children and to physically mature adults, because laboratory data suggest that individuals with strong attraction to both prepubertal children and physical mature adults are atypical [15]. However, we did include subjects who described themselves as sexually attracted to the adjacent groups, pubescents and physically mature adults. The full set of selection criteria, the only ones we ever formulated or applied to these data (J Krupp 2014, unpublished manuscript), were also applied to these data, without modification, in a subsequent study [27]. Table 1 shows the number of subjects after the foregoing exclusions.

**Table 2.** Numbers of heterosexual and homosexual subjects in one-son and two-son families.

| study | first and only son | | first of two sons | | second of two sons | |
|---|---|---|---|---|---|---|
| | heterosexual | homosexual | heterosexual | homosexual | heterosexual | homosexual |
| Blanchard & Bogaert [13] | 134 | 97 | 94 | 46 | 76 | 60 |
| Blanchard et al. (B-NB) [14] | 118 | 147 | 40 | 40 | 18 | 23 |
| Blanchard et al. (B-O) [14] | 59 | 87 | 27 | 32 | 36 | 54 |
| Blanchard et al. (H) [15] | 181 | 16 | 111 | 16 | 104 | 8 |
| Blanchard et al. (P) [15] | 41 | 24 | 34 | 11 | 15 | 22 |
| Blanchard et al. (T) [15] | 254 | 19 | 189 | 15 | 133 | 18 |
| Blanchard et al. [16] | 114 | 152 | 36 | 69 | 36 | 70 |
| Ellis & Blanchard [22] | 353 | 55 | 193 | 28 | 172 | 35 |
| Khorashad et al. [23] | 6 | 11 | 7 | 5 | 12 | 12 |
| Krupp (H) (J Krupp 2014, unpublished manuscript) | 47 | 14 | 14 | 7 | 10 | 7 |
| Krupp (P) (J Krupp 2014, unpublished manuscript) | 15 | 11 | 6 | 6 | 5 | 5 |
| Schagen et al. [25] | 419 | 27 | 178 | 23 | 160 | 27 |
| VanderLaan & Vasey [12] | 40 | 13 | 44 | 13 | 48 | 18 |
| VanderLaan et al. [26] | 88 | 140 | 52 | 75 | 35 | 78 |

## (b) Statistical analysis

In her mathematical proofs and in her practical suggestions for data analysis, Khovanova expressed the relative numbers of homosexual subjects as probabilities or proportions, that is, homosexual subjects/(homosexual subjects + heterosexual subjects). In the remainder of this article, we express the relative numbers of homosexual subjects as odds, that is, homosexual subjects/ heterosexual subjects. Correspondingly, where Khovanova expressed the magnitude and direction of effects as ratios of probabilities (i.e. risk ratios, relative risks), we use ratios of odds (which are simply called odds ratios). Our main reason for this was to make our results directly comparable to previous research on the FBOE, which has often used logistic regression, and logistic regression yields odds ratios rather than risk ratios.

In our terms, Khovanova's inequalities become $Odds_{22} > Odds_{12} > Odds_{11}$, and the effect estimates for the FBOE and the FFE become the odds ratios $Odds_{22}/Odds_{12}$ and $Odds_{12}/Odds_{11}$, respectively. The test for the inequality $Odds_{22} > Odds_{12}$ is the same as testing that $Odds_{22}/Odds_{12} > 1.00$, and the test for the inequality $Odds_{12} > Odds_{11}$ is the same as testing that $Odds_{12}/Odds_{11} > 1.00$. We conducted these tests using pooled estimates of the odds ratio obtained by meta-analysis of the 14 samples. We conducted one meta-analysis for the FBOE and one for the FFE.

The reason that we appended meta-analyses to Khovanova's basic procedure is that we expected the necessary matching procedure would result in substantial data loss and thus low statistical power for tests of individual samples. Both meta-analyses were performed with the Review Manager (RevMan) computer program [28]. Both used the inverse-variance weighting method and a random-effects model.

## 3. Results

The amount of data loss that resulted from selecting one-son and two-son families for analysis ranged from 12% to 68% in the 14 different samples (table 1). The median amount of data loss was about 30%.

The data used in the following meta-analyses are presented in table 2. Future researchers could simply add to the data in this table to update the meta-analyses with the results of additional studies.

Figure 1 shows the forest plot and inferential statistics for the meta-analysis of the FBOE. The outcome variable was the odds ratio $Odds_{22}/Odds_{12}$, that is, the odds that the second boy in a two-son family is homosexual divided by the odds that the first boy in a two-son family is homosexual. As we expected, most of the individual subsamples did not achieve statistical significance. The pooled odds ratio was 1.38, 95% CI [1.14, 1.66], which was significantly greater than the no-effect value of 1.00, $z = 3.35$, $p = 0.0008$. This means that the difference between zero and one older brother increased the odds of homosexuality by 38%. There was no evidence of heterogeneity among the samples. The statistical test for heterogeneity was not significant, $\chi^2_{13} = 12.81$, $p = 0.46$, and the quantitative estimate of inconsistency, $I^2$ [29] was 0%, suggesting that all of the variability in effect estimates could be attributed to sampling error.

A funnel plot for the FBOE odds ratios is presented in figure 2. A funnel plot is a scatterplot of the effect estimates from individual studies (here, the FBOE odds ratio) against a measure of each study's precision (here, the standard error of log FBOE odds ratio). In order to obtain the 'sides' of the funnel—the dotted lines in the figure—using the RevMan software, it was necessary to assume a fixed-effects model. An asymmetric 'funnel' (the typical shape of the distribution of data points) is usually interpreted as evidence of possible publication bias. Figure 2 shows no evidence of such asymmetry.

The forest plot and inferential statistics for the meta-analysis of the FFE are shown in figure 3. The outcome variable was the odds ratio $Odds_{12}/Odds_{11}$, that is, the odds that the first

| study or subgroup | second sons events | total | first sons events | total | weight | odds ratio IV, random, 95% CI |
|---|---|---|---|---|---|---|
| Blanchard & Bogaert [13] | 60 | 136 | 46 | 140 | 14.8% | 1.61 [0.99, 2.63] |
| Blanchard et al. (B-NB) [14] | 23 | 41 | 40 | 80 | 6.2% | 1.28 [0.60, 2.72] |
| Blanchard et al. (B-O) [14] | 54 | 90 | 32 | 59 | 8.0% | 1.27 [0.65, 2.46] |
| Blanchard et al. (H) [15] | 8 | 112 | 16 | 127 | 4.5% | 0.53 [0.22, 1.30] |
| Blanchard et al. (P) [15] | 22 | 37 | 11 | 45 | 4.0% | 4.53 [1.76, 11.66] |
| Blanchard et al. (T) [15] | 18 | 151 | 15 | 204 | 6.8% | 1.71 [0.83, 3.50] |
| Blanchard et al. [16] | 70 | 106 | 69 | 105 | 10.9% | 1.01 [0.57, 1.79] |
| Ellis & Blanchard [22] | 35 | 207 | 28 | 221 | 12.2% | 1.40 [0.82, 2.40] |
| Khorashad et al. [23] | 12 | 24 | 5 | 12 | 1.8% | 1.40 [0.35, 5.67] |
| Krupp (H) (J Krupp 2014, unpublished data) | 7 | 17 | 7 | 21 | 2.0% | 1.40 [0.37, 5.27] |
| Krupp (P) (J Krupp 2014, unpublished data) | 5 | 10 | 6 | 12 | 1.3% | 1.00 [0.19, 5.36] |
| Schagen et al. [25] | 27 | 187 | 23 | 201 | 10.0% | 1.31 [0.72, 2.37] |
| VanderLaan & Vasey [12] | 18 | 66 | 13 | 57 | 5.2% | 1.27 [0.56, 2.89] |
| VanderLaan et al. [26] | 78 | 113 | 75 | 127 | 12.4% | 1.55 [0.91, 2.63] |
| **total (95% CI)** | | **1297** | | **1411** | **100.0%** | **1.38 [1.14, 1.66]** |
| total events | 437 | | 386 | | | |

heterogeneity: $\tau^2 = 0.00$; $\chi^2 = 12.81$, d.f. = 13 ($p = 0.46$); $I^2 = 0\%$

test for overall effect: $Z = 3.35$ ($p = 0.0008$)

**Figure 1.** Forest plot and inferential statistics for the meta-analysis of the FBOE. 'Events' refers to homosexual subjects, and 'Total' refers to all subjects. The lozenge-shaped object at the bottom of the forest plot represents the pooled estimate of the odds ratio and its 95% confidence interval. See text for additional explanation. (Online version in colour.)

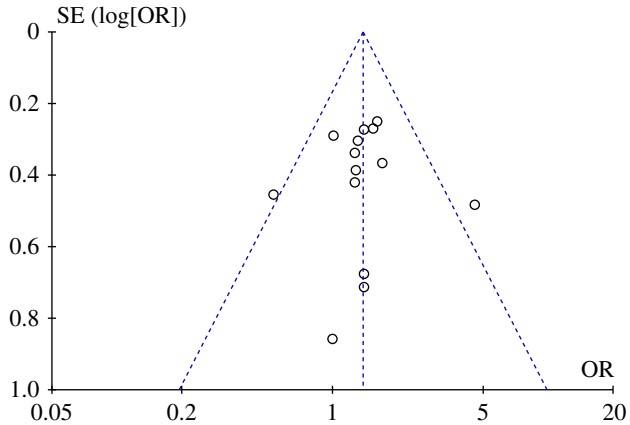

**Figure 2.** Funnel plot for the FBOE odds ratios. The triangle formed by the dashed lines represents a 95% confidence interval. (Online version in colour.)

boy in a two-son family is homosexual divided by the odds that the first (and only) boy in a one-son family is homosexual. The pooled odds ratio was 1.00, 95% CI [0.81, 1.23], which was obviously not different from the no-effect value of 1.00, $z = 0.00$, $p = 1.00$. This means that the difference between zero and one younger brother had no effect on the odds of homosexuality; in other words, the fecundity of homosexuals' mothers was the same as the fecundity of heterosexuals' mothers. There was again no evidence of heterogeneity among the samples. The statistical test for heterogeneity was not significant, $\chi^2_{13} = 18.52$, $p = 0.14$, and the quantitative estimate of inconsistency ($I^2$) was 30%, suggesting that variability in effect estimates could be attributed to sampling error.

Figure 4 shows a funnel plot for the FFE odds ratios. There was no notable relation between effect estimates and precision. Note that the estimates are less tightly clustered than are those for the FBOE (figure 2).

A series of secondary analyses examined the potential role of sampling bias in the present results. There was no evidence of a correlation between the amount of data lost from a sample and its estimate of the FBOE, $r = 0.10$, $p = 0.72$. By contrast, samples that lost more data yielded lower estimates of the FFE, $r = -0.52$, $p = 0.06$. A more fine-grained analysis examined the correlations of data loss with the odds of

homosexuality in only sons, first sons and second sons, that is, $Odds_{11}$, $Odds_{12}$ and $Odds_{22}$, respectively. The result for $Odds_{11}$ was $r = 0.04$, $p = 0.90$; that for $Odds_{12}$ was $r = -0.35$, $p = 0.22$; and that for $Odds_{22}$ was $r = -0.29$, $p = 0.32$. Thus, the correlation appears negligible for one-son families, but negative and similar in two-son families.

In a final analysis, we correlated the odds ratios for the FBOE and the FFE across the 14 samples. The Pearson correlation was $r = -0.45$, $p = 0.11$.

## 4. Discussion

As we anticipated, the selection of male subjects with zero or one older brothers (in other words, the selection of one-son and two-son families) resulted in substantial data loss—from 12% to 68% in the 14 samples. The amount of loss was lowest in samples from European countries (Germany, The Netherlands) and highest in samples from countries with more traditional cultures (Independent Samoa, Iran). We assume that this distinction results from differences in mean family size and not from differences in culture *per se*.

The median amount of data loss in our samples was about 30%. Our solution for restoring statistical power, which we planned from the start, was to meta-analyse the results from the individual samples. Other researchers might solve this problem by locating extremely large datasets that happen to include both sexual orientation and sibship data or else by explicitly recruiting heterosexual and homosexual subjects from one-son and two-son families.

Khovanova's method presupposes random sampling of one-son and two-son families, whereas our study selected one-son and two-son families from pre-existing convenience samples. Moreover, her language generally suggests that she is thinking in terms of sampling mothers, whereas the present datasets were assembled by sampling sons. Sampling mothers rather than sons would eliminate the relation between a sibship's size and the probability that one of its members will be chosen for a study [30]. Research on family demographics has previously been shown to be susceptible to complex, non-intuitive sampling artefacts that may be difficult to understand, let alone predict [31–33]. We, therefore, explored, as a purely

| study or subgroup | first sons events | total | only sons events | total | weight | odds ratio IV, random, 95% CI |
|---|---|---|---|---|---|---|
| Blanchard & Bogaert [13] | 46 | 140 | 97 | 231 | 11.8% | 0.68 [0.44, 1.05] |
| Blanchard et al. (B-NB) [14] | 40 | 80 | 147 | 265 | 10.2% | 0.80 [0.49, 1.32] |
| Blanchard et al. (B-O) [14] | 32 | 59 | 87 | 146 | 7.9% | 0.80 [0.44, 1.48] |
| Blanchard et al. (H) [15] | 16 | 127 | 16 | 197 | 6.0% | 1.63 [0.78, 3.39] |
| Blanchard et al. (P) [15] | 11 | 45 | 24 | 65 | 4.8% | 0.55 [0.24, 1.29] |
| Blanchard et al. (T) [15] | 15 | 204 | 19 | 273 | 6.4% | 1.06 [0.53, 2.14] |
| Blanchard et al. [16] | 69 | 105 | 152 | 266 | 10.9% | 1.44 [0.90, 2.30] |
| Ellis & Blanchard [22] | 28 | 221 | 55 | 408 | 10.5% | 0.93 [0.57, 1.52] |
| Khorashad et al. [23] | 5 | 12 | 11 | 17 | 1.7% | 0.39 [0.09, 1.78] |
| Krupp (H) (J Krupp 2014, unpublished data) | 7 | 21 | 14 | 61 | 3.1% | 1.68 [0.57, 4.97] |
| Krupp (P) (J Krupp 2014, unpublished data) | 6 | 12 | 11 | 26 | 2.1% | 1.36 [0.35, 5.38] |
| Schagen et al. [25] | 23 | 201 | 27 | 446 | 8.4% | 2.01 [1.12, 3.59] |
| VanderLaan & Vasey [12] | 13 | 57 | 13 | 53 | 4.5% | 0.91 [0.38, 2.19] |
| VanderLaan et al. [26] | 75 | 127 | 140 | 228 | 11.7% | 0.91 [0.58, 1.41] |
| **total (95% CI)** | | **1411** | | **2682** | **100.0%** | **1.00 [0.81, 1.23]** |
| total events | 386 | | 813 | | | |

heterogeneity: $\tau^2 = 0.04$; $\chi^2 = 18.52$, d.f. = 13 ($p = 0.14$); $I^2 = 30\%$
test for overall effect: Z = 0.00 ($p = 1.00$)

(odds ratio IV, random, 95% CI forest plot; x-axis 0.05, 0.2, 1, 5, 20; "higher in only sons" — "higher in first sons")

**Figure 3.** Forest plot and inferential statistics for the meta-analysis of the FFE. 'Events' refers to homosexual subjects, and 'Total' refers to all subjects. The lozenge-shaped object at the bottom of the forest plot represents the pooled estimate of the odds ratio and its 95% confidence interval. See text for additional explanation. (Online version in colour.)

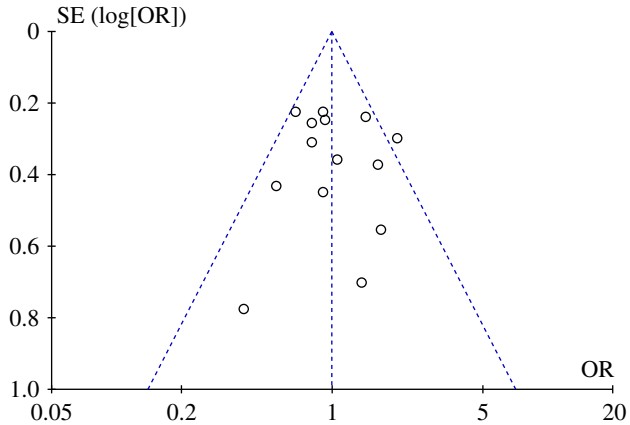

**Figure 4.** Funnel plot for the FFE odds ratios. The triangle formed by the dashed lines represents a 95% confidence interval. (Online version in colour.)

empirical matter, whether per cent of data loss correlates with estimates of the FBOE or FFE. There was no evidence of a correlation between the amount of data lost from a sample and its estimate of the FBOE. By contrast, samples that lost more data yielded lower estimates of the FFE, $r = -0.52$, $p = 0.06$ (two-tailed). This is difficult to explain with any single process, because data loss was associated with lower odds of homosexuality in two-son families but not in one-son families. Thus, we presently have no explanation of how the negative correlation between data loss and observed FFE arose. Nevertheless, the possibility that this finding is real and reproducible should be kept in mind when interpreting the study's results. If the effect is real, our observed correlation is probably a maximum estimate of its magnitude, because our samples varied greatly in mean family size.

We observed another result from Khovanova's method, which we had also not anticipated. The facts that the FBOE is estimated from subjects matched on female fecundity and the FFE is estimated from subjects matched on fraternal birth order do not mean that the computed estimates are themselves uncorrelated. Across the 14 samples, we found a negative correlation of $-0.45$ between the odds ratios for the FBOE and the odds ratios for the FFE. There are at least three possible explanations of this finding. The first is that it is simply the result of

random error. This is plausible, because the $p$-value for this correlation (0.11) was statistically non-significant, even without any correction for multiple comparisons.

The second possibility is that it is a statistical artefact, that Khovanova's method will always lead to a negative correlation between the estimated FBOE and estimated FFE. One might reason, for example, that a methodological artefact will arise because the variable $Odds_{12}$, the odds that the first boy in a two-son family is homosexual, serves as the numerator in the equation for the FFE and as the denominator in the equation for the FBOE. Thus, as the value of $Odds_{12}$ increases, the value of the FBOE will tend to decrease and the value of the FFE will tend to increase; as the value of $Odds_{12}$ decreases, the opposite will occur. If the other values in the equations, $Odds_{11}$ and $Odds_{22}$, were fixed, a negative correlation between FBOE and FFE would inevitably ensue. The problem with this hypothesis is that $Odds_{11}$ and $Odds_{22}$ are far from fixed; they, along with $Odds_{12}$, correlate highly with the proportion of homosexual subjects in a sample (see electronic supplementary material). If there are any cryptic causes of an inevitable correlation between the FBOE and the FFE, they might have to be inferred from simulated data rather than identified logically, and that is beyond the scope of this study.

The third possibility is that it is a genuine empirical result rather than a statistical artefact. In that case, one might interpret the negative correlation to mean that different samples contain different proportions of homosexual men whose sexual orientation derives from genetics or from the FBOE, and that the more there are of one type, the fewer there are of the other. In fact, one could expect natural samples to vary in this way. We cannot be sure, without actual biological data, whether this explanation is correct, but it is both plausible and intuitive. If there are two main aetiologies for homosexuality in men, one directly related to genes and one related to maternal immune responses to Y-linked antigens, then the more cases of one type there are in a given sample, the fewer cases of the other type there can be in that same sample.

Our first meta-analysis reproduced the FBOE, albeit mostly in samples that had shown the effect before. Of greater interest is the actual magnitude of the FBOE odds ratio, 1.38, which means that an increase from zero to one older brother is associated with a 38% increase in the odds of homosexuality.

The 95% confidence interval around our estimate, 1.14–1.66, contains 11 of the 13 odds ratio estimates reported by previous studies that calculated their estimates using logistic regression set-ups: 1.08 [34], 1.15 [35], 1.16 [36], 1.21 [37], 1.28 [22], 1.33 [13], 1.34 [12], 1.37 [38], 1.40 [39], 1.43 [40], 1.47 [25], 1.48 [16] and 1.68 [23]. It, therefore, appears that the estimate of 1.38 that we obtained with Khovanova's highly simplified model is similar in magnitude to estimates from studies that used families of unrestricted size and that counted sisters rather than ignoring them.

Our second meta-analysis produced no evidence of an FFE. The estimated odds ratio, 1.00, is about as close to zero effect as one can get. It is noteworthy that study [12], which we specifically included because the original analysis indicated both an FFE and an FBOE effect, looked similar to other samples (figures 1 and 3).

It is more difficult to compare the present FFE finding with previous FFE findings than it was to compare our FBOE finding with previous findings. Previous studies relevant to the FFE have used one or occasionally both of two very different research designs. The first design resembles a pedigree study as conducted in the field of genetics. In this case, the inherited trait of interest is fecundity. Homosexual and heterosexual subjects are compared with regard to the number of offspring produced by various classes of relatives, for example, mothers, grandmothers, sisters, brothers, aunts, uncles and cousins. Several such studies have found evidence of greater fecundity in the relatives of homosexual subjects, but they differ in finding this primarily on the mother's side [6–8,41–43], on the father's side [36,38] or about equally on both sides [44,45]. These studies might be useful for investigating patterns of transmission of fecundity-promoting genes (e.g. X-linkage), but comparing their outcomes to our quantitative estimate of the FFE is problematic.

The second design, which resembles Khovanova's method, was introduced by Camperio-Ciani *et al.* [6]. In this method, as in Khovanova's, one controls for fraternal birth order by comparing the numbers of younger siblings of firstborn heterosexual and homosexual men. It differs from Khovanova's method in that the subjects are firstborn *children*, not firstborn sons, female siblings are counted rather than ignored, and family size is not restricted in any way. Studies using this method have conducted statistical testing with Mann–Whitney or *F*-tests rather than logistic regression, so they did not generate odds ratios that could be compared directly with the result of the present study. These studies have found evidence of higher fertility in homosexuals' mothers in two samples [8,46], no significant difference between the mothers of homosexual and heterosexual males in four samples [6,7,47], and lower fertility in homosexuals' mothers in four samples [47]. (The number of samples is greater than the number of studies because study [47] included six samples.)

If one is willing to assume, in the absence of a formal meta-analysis, that the contradictory findings obtained with the Camperio-Ciani *et al* [6] method probably signal no difference in fertility between the mothers of firstborn homosexual and firstborn heterosexual males, then the null finding we obtained with the similar Khovanova method would be consistent with the previous research. Even this conditional conclusion is qualified, however, by our finding that the Khovanova method, when applied to archival convenience samples, may introduce method artefacts that artificially lower estimates of the FFE, especially those computed on high-fertility populations.

## 5. Summary and conclusion

The goals of Khovanova's method were to produce quantitative estimates of the FBOE that would not be affected by the FFE and quantitative estimates of the FFE that would not be affected by the FBOE. The goal of the present study was to investigate the performance of Khovanova's method when applied to real data from pre-existing datasets.

The formula for estimating the FBOE yielded an odds ratio that was completely typical of odds ratios previously obtained using a different approach, namely, logistic regression analysis. This is remarkable in that these regression analyses did not exclude sisters or limit family size in any way. The Khovanova and regression methods converge on the estimate that each older brother increases the odds of homosexuality in later-born males by about 30–40%. All the results concerning the FBOE point to the conclusion that Khovanova's method can be used to estimate this parameter in low-fecundity populations as well as in high-fecundity populations.

The findings concerning Khovanova's estimate of the FFE are more difficult to evaluate or interpret. The findings of the most methodologically comparable previous studies have ranged from higher fecundity in the mothers of homosexuals to higher fecundity in the mothers of heterosexuals. It is, therefore, arbitrary to say whether the present estimate of the FFE—which was equivalent to no fecundity effect at all—is consistent with such prior research.

There are at least two other issues with Khovanova's method for estimating the FFE. The first is that the amount of data lost from individual samples by selecting one-son and two-son families correlated with the estimated FFE. This suggests that the use of Khovanova's procedure to estimate the FFE from pre-existing datasets may have led to biased results. The second concerns Khovanova's assumption that female offspring can safely be ignored in the interests of simplicity and clarity. The FBOE is specified in terms of male offspring; ignoring females can be justified on theoretical grounds, and the present quantitative results support that simplifying assumption. The FFE, on the other hand, is specified in terms of numbers of offspring, not numbers of sons. This raises the question whether the quantity measured by Khovanova's formula for the FFE comports with researchers' concepts of 'fecundity.' For the foregoing reasons, further research must decide whether Khovanova's method is as suitable for estimating an FFE independent of any FBOE as it is for estimating an FBOE independent of any FFE. Until that is known, it seems desirable to study the FFE by counting the offspring of homosexual men's sisters, maternal aunts, maternal grandmothers and so on, in addition to analysing the offspring of their own mothers.

**Ethics.** This study was carried out on archived data. Institutional ethics approvals were obtained by the original investigators.

**Data accessibility.** All meta-analysed data are included in the text.

**Authors' contributions.** R.B. carried out statistical analyses and wrote the first draft of the manuscript. K.J.Z. contributed material to subsequent drafts and edited all drafts. J.K. created a dataset of unpublished material from a larger project. D.P.V. and P.L.V. collected the original data for one sample and retrieved data from it for this project.

**Competing interests.** The authors declare no competing interests.

**Funding.** No funding was sought or needed for this study.

**Acknowledgements.** The authors thank Klaus M. Beier and Dorit Grundmann for helping them locate an unpublished dataset.

## Endnotes

[1]We use the word *homosexuality* in this article rather than the increasingly popular word *androphilia*, because homosexuality simply denotes the erotic preference for members of one's own biological sex, whereas androphilia denotes the erotic preference for *physically mature* males. Two of our study groups erotically preferred boys in Tanner Stage 1 of pubertal development, and another two groups erotically preferred boys in Tanner Stages 2–3. These are accurately (and traditionally) called homosexual paedophiles and homosexual hebephiles, respectively, but it would be self-contradictory to call them androphilic paedophiles or androphilic hebephiles. In contrast, the word *homosexual* applies equally to all our same-sex attracted groups.

[2]It would have been more precise to use the term *maternal fertility* in this article rather than *female fecundity*. The only females we are concerned with are the mothers of the index subjects, and we are concerned with the number of children a woman actually produces rather than a woman's physiological capacity for reproduction. However, the author of the article foundational to this one [1] used the term maternal fecundity, so we have used the same language for the greater ease of people who might read Khovanova's article along with ours.

[3]The statistic has the name *risk ratio* for historical reasons; it can be computed for good things, bad things, or neutral things.

[4]Khovanova herself does not give this alternative interpretation of the ratio $p_{12}/p_{11}$. It is implied, however, by restricting the universe of discourse to boys.

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
