## [Reviewer comments · Proceedings of the Royal Society B: Biological Sciences]

Review History

RSPB-2019-2907.R0 (Original submission)

Review form: Reviewer 1

Recommendation

Accept with minor revision (please list in comments)

Scientific importance: Is the manuscript an original and important contribution to its field?

Excellent

General interest: Is the paper of sufficient general interest?

Good

Quality of the paper: Is the overall quality of the paper suitable?

Excellent

Is the length of the paper justified?

Yes

Should the paper be seen by a specialist statistical reviewer?

No

Do you have any concerns about statistical analyses in this paper? If so, please specify them explicitly in your report.

No

It is a condition of publication that authors make their supporting data, code and materials available - either as supplementary material or hosted in an external repository. Please rate, if applicable, the supporting data on the following criteria.

Is it accessible?

Yes

Is it clear?

Yes

Is it adequate?

Yes

Do you have any ethical concerns with this paper?

No

Comments to the Author

The issues of causation and evolutionary persistence of male homosexuality have fascinated both scientists and laypersons for decades. Recently there has been progress in both. Blanchard has established the fact (at least as much as any scientific claim can be a fact) of the fraternal birth order effect (FBOE), which is the single most powerful etiological factor known for male homosexuality. (His maternal immune hypothesis is fascinating, and I hope it's true, but it's irrelevant to this manuscript.) With respect to evolutionary hypotheses, researchers have begun to collect relevant data, including data from relatively traditional societies likely to be more similar than the contemporary West to the EEA. One strain of this work looks at the fecundity of mothers of homosexual men: the female fecundity effect (FFE). However, the latter literature is certainly limited, and increasingly so, by the modernized constraints on fertility–birth control and family planning. (I suspect this is true even in Samoa and other non-Western cultures studied.)

Larger families produce more sons, and this will increase the FBOE among them. There is, then, a confounding across families between the FFE and the FBOE. This is inconvenient, especially for researchers studying whether male homosexuality is associated with larger family size for evolutionary reasons.

A recent article by Khovanova provides a very useful way to distinguish—or at least to investigate independently—the FBOE and the FFE. One contribution of Khovanova's method that I don't recall being highlighted in the present manuscript is that it can be applied as well in populations with low fecundity as in those with high fecundity.

The present article applies Khovanova's insights across 14 samples and finds persuasive evidence for the FBOE—as it should, given its highly replicable nature—but not for the FFE. The results are very useful both as an illustration and as a substantive contribution. I found this article elegant, interesting, and important. And cool.

One touchy issue I'll raise is the exclusion of subjects from the Krupp sample who are attracted to both children and adults. As the authors undoubtedly know, exclusions have been misused by researchers in p-hacking for significance. I do not accuse the authors of doing this, but suggest that they reassure the reader that this decision is just like what they've done elsewhere or makes no difference.

The negative correlation between the FBOE and FFE is interesting. (BTW, what was the p value

for the correlation?) I do not demand clarification of the following issue, but its clarification would be useful if possible: is this an empirical or an analytic result? That is, is there something about the arithmetic/mathematics of the method that will always produce such a correlation, or is there something potentially interesting and informative that this was found? One way to look at this would be to simulate data randomly with respect to the sexual orientation of first- and second-born brothers and examine whether one gets a non-zero correlation. But again, this is not necessary to satisfy this reviewer.

Review form: Reviewer 2

Recommendation

Accept as is

Scientific importance: Is the manuscript an original and important contribution to its field?

Excellent

General interest: Is the paper of sufficient general interest?

Excellent

Quality of the paper: Is the overall quality of the paper suitable?

Excellent

Is the length of the paper justified?

Yes

Should the paper be seen by a specialist statistical reviewer?

No

Do you have any concerns about statistical analyses in this paper? If so, please specify them explicitly in your report.

No

It is a condition of publication that authors make their supporting data, code and materials available - either as supplementary material or hosted in an external repository. Please rate, if applicable, the supporting data on the following criteria.

Is it accessible?

Yes

Is it clear?

Yes

Is it adequate?

Yes

Do you have any ethical concerns with this paper?

No

Comments to the Author

The present paper aimed to apply the Khovanova's method of producing quantitative estimates of the FBOE that would not be affected by the FFE and quantitative estimates of the FFE that would not be affected by the FBOE to real data from pre-existing datasets. The study is well-

executed, and the paper is well written. More importantly, it provides important new data that enable the understanding of an important phenomenon. On this basis, I favor publication.

Review form: Reviewer 2

Recommendation

Major revision is needed (please make suggestions in comments)

Scientific importance: Is the manuscript an original and important contribution to its field?

Acceptable

General interest: Is the paper of sufficient general interest?

Acceptable

Quality of the paper: Is the overall quality of the paper suitable?

Good

Is the length of the paper justified?

Yes

Should the paper be seen by a specialist statistical reviewer?

No

Do you have any concerns about statistical analyses in this paper? If so, please specify them explicitly in your report.

Yes

It is a condition of publication that authors make their supporting data, code and materials available - either as supplementary material or hosted in an external repository. Please rate, if applicable, the supporting data on the following criteria.

Is it accessible?

Yes

Is it clear?

Yes

Is it adequate?

N/A

Do you have any ethical concerns with this paper?

No

Comments to the Author

Comments are in the attached file. (See Appendix A)

Decision letter (RSPB-2019-2907.R0)

07-Feb-2020

Dear Dr Blanchard:

Your manuscript has now been peer reviewed and the reviews have been assessed by an Associate Editor. The reviewers' comments (not including confidential comments to the Editor) and the comments from the Associate Editor are included at the end of this email for your reference. As you will see, the reviewers and the Editors have raised some concerns with your manuscript and we would like to invite you to revise your manuscript to address them.

Research ethics:

Use of animals and field studies:

Please submit a copy of your revised paper within three weeks. If we do not hear from you within this time your manuscript will be rejected. If you are unable to meet this deadline please let us know as soon as possible, as we may be able to grant a short extension.

Best wishes,

Dr Locke Rowe

Associate Editor

Comments to Author:

We have now heard from three experts in the field. All three like your study, but Reviewer 3 has raised some concerns that you need to address before we move forward. I am recommending that you revise manuscript in the light of these concerns -- and respond to the comments of the reviewers, particularly those of Reviewer 3.

Reviewer(s)' Comments to Author:

Referee: 1

Comments to the Author(s)

The issues of causation and evolutionary persistence of male homosexuality have fascinated both scientists and laypersons for decades. Recently there has been progress in both. Blanchard has established the fact (at least as much as any scientific claim can be a fact) of the fraternal birth order effect (FBOE), which is the single most powerful etiological factor known for male homosexuality. (His maternal immune hypothesis is fascinating, and I hope it's true, but it's

irrelevant to this manuscript.) With respect to evolutionary hypotheses, researchers have begun to collect relevant data, including data from relatively traditional societies likely to be more similar than the contemporary West to the EEA. One strain of this work looks at the fecundity of mothers of homosexual men: the female fecundity effect (FFE). However, the latter literature is certainly limited, and increasingly so, by the modernized constraints on fertility–birth control and family planning. (I suspect this is true even in Samoa and other non-Western cultures studied.)

Larger families produce more sons, and this will increase the FBOE among them. There is, then, a confounding across families between the FFE and the FBOE. This is inconvenient, especially for researchers studying whether male homosexuality is associated with larger family size for evolutionary reasons.

A recent article by Khovanova provides a very useful way to distinguish—or at least to investigate independently—the FBOE and the FFE. One contribution of Khovanova’s method that I don’t recall being highlighted in the present manuscript is that it can be applied as well in populations with low fecundity as in those with high fecundity.

The present article applies Khovanova’s insights across 14 samples and finds persuasive evidence for the FBOE—as it should, given its highly replicable nature—but not for the FFE. The results are very useful both as an illustration and as a substantive contribution. I found this article elegant, interesting, and important. And cool.

One touchy issue I’ll raise is the exclusion of subjects from the Krupp sample who are attracted to both children and adults. As the authors undoubtedly know, exclusions have been misused by researchers in p-hacking for significance. I do not accuse the authors of doing this, but suggest that they reassure the reader that this decision is just like what they’ve done elsewhere or makes no difference.

The negative correlation between the FBOE and FFE is interesting. (BTW, what was the p value for the correlation?) I do not demand clarification of the following issue, but its clarification would be useful if possible: is this an empirical or an analytic result? That is, is there something about the arithmetic/mathematics of the method that will always produce such a correlation, or is there something potentially interesting and informative that this was found? One way to look at this would be to simulate data randomly with respect to the sexual orientation of first- and second-born brothers and examine whether one gets a non-zero correlation. But again, this is not necessary to satisfy this reviewer.

Referee: 2

Comments to the Author(s)

The present paper aimed to apply the Khovanova’s method of producing quantitative estimates of the FBOE that would not be affected by the FFE and quantitative estimates of the FFE that would not be affected by the FBOE to real data from pre-existing datasets. The study is well-executed, and the paper is well written. More importantly, it provides important new data that enable the understanding of an important phenomenon. On this basis, I favor publication.

Referee: 3

Comments to the Author(s)

Comments are in the attached file.

Author's Response to Decision Letter for (RSPB-2019-2907.R0)

See Appendix B.

Decision letter (RSPB-2019-2907.R1)

26-Feb-2020

Dear Dr Blanchard

I am pleased to inform you that your manuscript entitled "A Method Yielding Comparable Estimates of the Fraternal Birth Order and Female Fecundity Effects in Male Homosexuality" has been accepted for publication in Proceedings B.

Open Access

You are invited to opt for Open Access, making your freely available to all as soon as it is ready for publication under a CCBY licence. Our article processing charge for Open Access is £1700. Corresponding authors from member institutions (<http://royalsocietypublishing.org/site/librarians/allmembers.xhtml>) receive a 25% discount to these charges. For more information please visit <http://royalsocietypublishing.org/open-access>.

Paper charges

Sincerely,

Dr Locke Rowe

Associate Editor:

Board Member

Comments to Author:

Thank you for submitting your revised manuscript to Proceedings of the Royal Society B. I am pleased to recommend acceptance. Congratulations on a fine paper.

Appendix A

1. The result is that the FBOE is confirmed, but not the FFE. Thus the title, referring to « comparables estimates » of the FBOE and FFE, is quite misleading and seems contradictory with the results.
2. Page 13. The fact that individuals, rather than their mother, are sampled, does not eliminate the relation between a sibship's size and the probability that one of its members will be chosen for a study. This is mentioned in the discussion, but not really discussed. Is there a link between that point and the observed correlation between both estimates ?
3. Page 10. « *We conducted one meta-analysis for the FBOE and one for the FFE.* » Thus the estimates for the FBOE and FFE are not concomitantly estimated. This is problematic as there is a correlation between both estimates (-0.45 , page 12, see also point 4)
4. Page 12. « *The p-value for this correlation is of no interest because it is beside the point of this investigation, namely, to show there is no mathematically necessary zero correlation between the two estimated odds ratios* » Please provide this P-value. The correlation between the two estimates is of interest in the context of this manuscript.
5. Page 13. « *.. samples that lost more data yielded lower estimates of the FFE, $r = -.52$.* » . This suggests that FFE would be better estimated from a full dataset, which is impossible using Khovanova's method. It is thus possible that the FFE is not captured by this analysis restricted to families with only one or two sons. It is thus pivotal to present a power analysis for this negative result: what is the probability to find a significant result, considering this dataset, under the alternative hypothesis that there is a FFE ?
 - 5a. If the power calculated (see point 5) is low, then there is not much to say concerning FFE with that dataset... and the manuscript is restricted to a simple meta analysis of the FBOE controlling for FFE. The added value of this meta-analysis is questionable, as it is said page 14, that "*Our first meta-analysis reproduced the FBOE, albeit mostly in samples that had shown the effect before*". For a solid meta-analysis of the FBOE, more studies are probably to be considered, as was already done previously (see e.g. Blanchard 2018a and 2018b, cited in the paper).
6. Page 7. "*Khovanova's procedure is simple, logical, and elegant. It is certainly more transparent than multivariate procedures for disentangling the effects of birth order and family size.*" This is perhaps true, but considering that Khovanova's method is associated with a mean of 30% data loss (up to 68% here), which method is preferable is not obvious. Perhaps Khovanova's method is not the best for FFE for the data set presented here (see point 5). A comparison with other published methods should be presented rather than this assertion not backed up with evidence. For example, how Khovanova's method performs compared to (classical) regression procedures able to study FFE while controlling FBOE (and reciprocally) ? This is very simple to do with the data set presented in this study.
7. What about the method proposed by Blanchard in Arch Sex Behav (2014) 43:845–852 ? (title: *Detecting and Correcting for Family Size Differences in the Study of Sexual Orientation and Fraternal Birth Order*). Could it be also compared with Khovanova's method ?
8. Acronyms "FBOE" and "FFE" are used in the abstract but are defined later in the introduction.

Appendix B

Dear Dr. Rowe,

We are pleased to have the opportunity to respond to the referees' comments on my manuscript, "A Method Yielding Comparable Estimates of the Fraternal Birth Order and Female Fecundity Effects in Male Homosexuality." We found the referees' comments helpful, and we think that responding to them has improved the revised manuscript. We have tried to make additions to the manuscript, requested by the reviewers, as concise as possible.

A copy of the manuscript with tracked changes is appended to this letter.

Please note that we have changed the format of the references to numbered references in the clean copy of the revised manuscript but not in the tracked changes version.

REFEREE 1

First point: One contribution of Khovanova's method that I don't recall being highlighted in the present manuscript is that it can be applied as well in populations with low fecundity as in those with high fecundity.

We have added the following sentence to the section headed, Summary and Conclusions: **"All the results concerning the FBOE point to the conclusion that Khovanova's method can be used to estimate this parameter in low-fecundity populations as well as in high-fecundity populations."**

Second point: One touchy issue I'll raise is the exclusion of subjects from the Krupp sample who are attracted to both children and adults. As the authors undoubtedly know, exclusions have been misused by researchers in p-hacking for significance. I do not accuse the authors of doing this, but suggest that they reassure the reader that this decision is just like what they've done elsewhere or makes no difference.

We have added the material shown in boldface: "In the present study we excluded self-described bisexuals, because they did not fit into the research design and because we had no hypothesis about them, and we excluded teleiophiles, because hardly any of the teleiophiles were homosexual. **We also excluded subjects who described themselves as attracted both to prepubertal children and to physically mature adults, because laboratory data suggest that individuals with strong attraction to both prepubertal children and physical mature adults are atypical (see Blanchard et al., 2012, Figure 1). However, we did include subjects who described themselves as sexually attracted to the adjacent groups, pubescents and physically mature adults. The full set of selection criteria, the only ones we ever formulated or applied to Krupp's data, were also applied to these data, without modification, in a subsequent study by Blanchard et al. (2020). Table 1 shows the number of subjects after the foregoing exclusions.**"

Third point: The negative correlation between the FBOE and FFE is interesting. (BTW, what was the p value for the correlation?) I do not demand clarification of the following issue, but its clarification would be useful if possible: is this an empirical or an analytic result? That is, is

there something about the arithmetic/mathematics of the method that will always produce such a correlation, or is there something potentially interesting and informative that this was found?

We have now added the p -value in the Results section as well as much more material in the Discussion section about the interpretation of this result.

“We observed another result from Khovanova’s method, which we had also not anticipated. The facts that the FBOE is estimated from subjects matched on female fecundity and the FFE is estimated from subjects matched on fraternal birth order do not mean that the computed estimates are themselves uncorrelated. Across the 14 samples, we found a negative correlation of $-.45$ between the odds ratios for the FBOE and the odds ratios for the FFE. There are at least three possible explanations of this finding. This first is that it is simply the result of random error. This is plausible, because the p -value for this correlation (.11) was statistically nonsignificant, even without any correction for multiple comparisons.

“The second possibility is that it is a statistical artifact, that Khovanova’s method will always lead to a negative correlation between the estimated FBOE and estimated FFE. One might reason, for example, that a methodological artifact will arise because the variable Odds₁₂, the odds that the first boy in a two-son family is homosexual, serves as the numerator in the equation for the FFE and as the denominator in the equation for the FBOE. Thus, as the value of Odds₁₂ increases, the value of the FBOE will tend to decrease and the value of the FFE will tend to increase; as the value of Odds₁₂ decreases, the opposite will occur. If the other values in the equations, Odds₁₁ and Odds₂₂, were fixed, a negative correlation between FBOE and FFE would inevitably ensue. The problem with this hypothesis is that Odds₁₁ and Odds₂₂ are far from fixed; they, along with Odds₁₂, correlate highly with the proportion of homosexual subjects in a sample. If there are any cryptic causes of an inevitable correlation between the FBOE and the FFE, they might have to be inferred from simulated data rather than identified logically, and that is beyond the scope of this study.

“The third possibility is that it is a genuine empirical result rather than a statistical artifact. In that case, one might interpret the negative correlation to mean that different samples contain different proportions of homosexual men whose sexual orientation derives from genetics or from the FBOE, and that the more there are of one type, the fewer there are of the other. In fact, one could expect natural samples to vary in this way. We cannot be sure, without actual biological data, whether this explanation is correct, but it is both plausible and intuitive. If there are two main etiologies for homosexuality in men, one directly related to genes and one related to maternal immune responses to Y-linked antigens, then the more cases of one type there are in a given sample, the fewer cases of the other type there can be in that same sample.”

REFEREE 2

This referee had no criticisms or suggestions.

REFEREE 3

1. The result is that the FBOE is confirmed, but not the FFE. Thus the title, referring to « comparables estimates » of the FBOE and FFE, is quite misleading and seems contradictory with the results.

The current title is “A Method Yielding Comparable Estimates of the Fraternal Birth Order and Female Fecundity Effects in Male Homosexuality.” The word “comparable” here means that the estimates were computed in a similar manner and that the obtained values are expressed in the same metric (in this case, as odds ratios). It does not mean that the empirically observed estimates were similar in magnitude.

We think that that meaning is made clear in the second sentence of the abstract: “In a recent paper, T. Khovanova proposed **a novel method for computing independent estimates of these effects on the same samples and expressing the magnitude and direction of the effects in the same metric.**”

2. Page 13. The fact that individuals, rather than their mother, are sampled, does not eliminate the relation between a sibship’s size and the probability that one of its members will be chosen for a study. This is mentioned in the discussion, but not really discussed. Is there a link between that point and the observed correlation between both estimates?

There seem to be two different points here. We don’t understand the first one. We have not argued that sampling probands rather than mothers eliminates the relation between sibship size and the probability of a proband being sampled. In fact, we argued the opposite, “**Sampling mothers rather than sons would eliminate the relation between a sibship’s size and the probability that one of its members will be chosen for a study (see Bytheway, 1974).**”

Our response to the second point is that we do not see any relation between sibship size and the probability of a proband being sampled, on the one hand, and the observed correlation between the FBOE and the FFE, on the other.

3. Page 10. « We conducted one meta-analysis for the FBOE and one for the FFE. » Thus the estimates for the FBOE and FFE are not concomitantly estimated. This is problematic as there is a correlation between both estimates (-0.45 , page 12, see also point 4).

In the first place, we do not understand the precise point that the referee is making or exactly what alternative procedure the referee thinks would be better than the one suggested by Khovanova. In our view, the main appeal of Khovanova’s procedure is that she proposes a way to avoid the confounding of the FBOE and the FFE by means of experimental design. Experimental-design approaches are generally better than trying to correct confounds after the fact by means of covariance or covariance-like procedures.

In the second place, the referee assumes that the negative correlation between the FBOE and FFE estimates is a problem, a matter to be corrected. There is not the slightest basis for this assumption. It is quite possible that this correlation is a genuine finding that reflects something in nature. We did not, unfortunately, make this point in the first version of the manuscript, but we

have stated it explicitly in the revision: **“The third possibility is that it is a genuine empirical result rather than a statistical artifact. In that case, one might interpret the negative correlation to mean that different samples contain different proportions of homosexual men whose sexual orientation derives from genetics or from the FBOE, and that the more there are of one type, the fewer there are of the other. In fact, one could expect natural samples to vary in this way. We cannot be sure, without actual biological data, whether this explanation is correct, but it is both plausible and intuitive. If there are two main etiologies for homosexuality in men, one directly related to genes and one related to maternal immune responses to Y-linked antigens, then the more cases of one type there are in a given sample, the fewer cases of the other type there can be in that same sample.”**

4. Page 12. *« The p-value for this correlation is of no interest because it is beside the point of this investigation, namely, to show there is no mathematically necessary zero correlation between the two estimated odds ratios » Please provide this P-value. The correlation between the two estimates is of interest in the context of this manuscript.*

The referee is absolutely right. We have now provided the *p*-value and discussed the possible explanations of the negative correlation between the FBOE and the FFE in detail. We have already quoted the relevant additions.

5. Page 13. *« .. samples that lost more data yielded lower estimates of the FFE, $r = -.52$. » . This suggests that FFE would be better estimated from a full dataset, which is impossible using Khovanova’s method. It is thus possible that the FFE is not captured by this analysis restricted to families with only one or two sons. It is thus pivotal to present a power analysis for this negative result: what is the probability to find a significant result, considering this dataset, under the alternative hypothesis that there is a FFE ?*

The referee’s points 5, 5a, and 6 are somewhat overlapping and seem to reflect a general concern that Khovanova’s method is an unfair or inadequate approach to estimating the FFE. We think it is appropriate to address this general concern before responding to more specific suggestions and criticisms made by the referee.

We have made every effort to seek out, and to report, all possible problems with Khovanova’s suggested procedure. We feel that if we have erred at all, it may be on the side of excess caution. Here are the various caveats we have included in the manuscript:

“Thus, we presently have no explanation of how the negative correlation between data loss and observed FFE arose. Nevertheless, the possibility that this finding is real and reproducible should be kept in mind when interpreting the study’s results. If the effect is real, our observed correlation is probably a maximum estimate of its magnitude, because our samples varied greatly in mean family size.”

“... the Khovanova method, when applied to archival convenience samples, may introduce method artifacts that artificially lower estimates of the FFE, especially those computed on high-fertility populations.”

“There are at least two other issues with Khovanova’s method for estimating the FFE. The first is that the amount of data lost from individual samples by selecting one-son and two-son families correlated with the estimated FFE. This suggests that the use of Khovanova’s procedure to estimate the FFE from pre-existing datasets may have led to biased results. The second concerns Khovanova’s assumption that female offspring can safely be ignored in the interests of simplicity and clarity. The FBOE is specified in terms of male offspring; ignoring females can be justified on theoretical grounds, and the present quantitative results support that simplifying assumption. The FFE, on the other hand, is specified in terms of numbers of offspring, not numbers of sons. This raises the question whether the quantity measured by Khovanova’s formula for the FFE comports with researchers’ concepts of “fecundity.” For the foregoing reasons, further research must decide whether Khovanova’s method is as suitable for estimating an FFE independent of any FBOE as it is for estimating an FBOE independent of any FFE. Until that is known, it seems desirable to study the FFE by counting the offspring of homosexual men’s sisters, maternal aunts, maternal grandmothers, and so on, in addition to analyzing the offspring of their own mothers.”

With regard to specific matters from the referee’s point 5, we believe that a power analysis for Khovanova’s method - however the referee thinks this would be implemented in reality – is impossible and unnecessary. A power analysis requires a prior estimate of effect size. There is no source for such an estimate except the present study, and that estimate is an effect size of zero. Note that this estimate is not inconsistent with the aggregate of prior studies of primiparous mothers of homosexual men, a topic reviewed in the manuscript. It is possible that primiparous mothers are not the best way of studying the FFE. That point is related to the last sentence in the manuscript: **“it seems desirable to study the FFE by counting the offspring of homosexual men’s sisters, maternal aunts, maternal grandmothers, and so on, in addition to analyzing the offspring of their own mothers.”**

5a. If the power calculated (see point 5) is low, then there is not much to say concerning FFE with that dataset... and the manuscript is restricted to a simple meta analysis of the FBOE controlling for FFE. The added value of this meta-analysis is questionable, as it is said page 14, that “Our first meta-analysis reproduced the FBOE, albeit mostly in samples that had shown the effect before”. For a solid meta-analysis of the FBOE, more studies are probably to be considered, as was already done previously (see e.g. Banchard 2018a and 2018b, cited in the paper).

We are unsure where the referee is going with this, that is, whether the referee thinks we should not report our findings for the FFE or not report any of our findings at all. Obviously, we vigorously disagree. Khovanova’s procedure for studying the FBOE and FFE was specifically designed to study these phenomena using a formal and explicit mathematical logic. It is clear that it produced the right results for the FBOE, and it is not at all clear that it produced the wrong results for the FFE.

It is conceivable that the FFE results produced by Khovanova’s method apply only to primiparous mothers of gay men. The question of whether primiparous mothers of gay men are

different from multiparous mothers of gay men has been raised before. This is a question that can be resolved only by further empirical study, not by suppressing the present findings.

6. Page 7. *“Khovanova’s procedure is simple, logical, and elegant. It is certainly more transparent than multivariate procedures for disentangling the effects of birth order and family size.” This is perhaps true, but considering that Khovanova’s method is associated with a mean of 30% data loss (up to 68% here), which method is preferable is not obvious. Perhaps Khovanova’s method is not the best for FFE for the data set presented here (see point 5). A comparison with other published methods should be presented rather than this assertion not backed up with evidence. For example, how Khovanova’s method performs compared to (classical) regression procedures able to study FFE while controlling FBOE (and reciprocally)? This is very simple to do with the data set presented in this study.*

The referee seems to be suggesting that we look for positive evidence of an FFE using an entirely different statistical approach and then compare the results of that analysis with the results obtained with Khovanova’s method. Presumably the referee is expecting or hoping that an alternative procedure will turn up positive evidence for the existence of an FFE and thus show that the conclusion following from Khovanova’s method is false.

We regard this exercise as entirely unnecessary. There already are published studies that use an experimental design similar to Khovanova’s procedure for estimating the FFE. We already reviewed this literature and compared it to Khovanova’s estimate of the FFE in the original version of the manuscript. This is the relevant material:

“The second design, which resembles Khovanova’s method, was introduced by Camperio-Ciani et al. (2004). In this method, as in Khovanova’s, one controls for fraternal birth order by comparing the numbers of younger siblings of firstborn heterosexual and homosexual men. It differs from Khovanova’s method in that the subjects are firstborn *children*, not firstborn sons, female siblings are counted rather than ignored, and family size is not restricted in any way. Studies using this method have conducted statistical testing with Mann-Whitney or *F*-tests rather than logistic regression, so they did not generate odds ratios that could be compared directly with the result of the present study. These studies have found evidence of higher fertility in homosexuals’ mothers in two samples (Iemmola & Camperio Ciani, 2009; Rieger et al., 2012), no significant difference between the mothers of homosexual and heterosexual males in four samples (Blanchard, 2012; Camperio Ciani et al., 2004, 2009), and lower fertility in homosexuals’ mothers in four samples (Blanchard, 2012). (The number of samples is greater than the number of studies because Blanchard (2012) studied six samples.)

“If one is willing to assume, in the absence of a formal meta-analysis, that the contradictory findings obtained with the Camperio-Ciani et al. (2004) method probably signal no difference in fertility between the mothers of firstborn homosexual and firstborn heterosexual males, then the null finding we obtained with the similar Khovanova method would be consistent with the previous research.”

Now, the referee might object that primiparous mothers of gay men could be different from parous mothers of gay men, and we would agree with him. It is possible that examining the fecundity of primiparous mothers is not the best way to study the FFE, and the concluding sentence of the revised manuscript alludes to this. However, the present study is not the place to go into this. It would change the nature of the project from a methodological study of a proposed novel approach to estimating the FBOE and FFE into an attempt to confirm a specific evolutionary psychology theory of homosexuality, and it would greatly lengthen the manuscript in the process.

7. What about the method proposed by Blanchard in Arch Sex Behav (2014) 43:845–852 ? (title: Detecting and Correcting for Family Size Differences in the Study of Sexual Orientation and Fraternal Birth Order). Could it be also compared with Khovanova's method ?

Blanchard (2014) does not address female fecundity/family size except as a nuisance variable. It cannot be compared to the present study.

8. Acronymes "FBOE" and "FFE" are used in the abstract but are defined later in the introduction.

The abbreviations FBOE and FFE are now defined in the first sentence of the Abstract.

Sincerely,

Ray Blanchard